# Rehydration Beverages Made from *Quercus sideroxyla* Infusion, Probiotics, and Prebiotics: Antioxidant and Anti-Inflammatory Potential

**DOI:** 10.3390/foods14050837

**Published:** 2025-02-28

**Authors:** Carlos Alonso Salas-Ramírez, Martha Rocío Moreno-Jiménez, Nuria Elizabeth Rocha-Guzmán, José Alberto Gallegos-Infante, Rubén Francisco González-Laredo, Silvia Marina González Herrera, Manuel Efraín González-Mercado, Karen Marlenne Herrera-Rocha, Manuel Humberto Cháirez-Ramirez

**Affiliations:** 1Laboratorio Nacional CONAHCYT de Apoyo a la Evaluación de Productos Bióticos (LaNAEPBi), Unidad de Servicio, Tecnológico Nacional de México/I TecNM/ITD, Blvd. de Durango, Felipe Pescador 1830 Ote., Durango 34080, Mexico; saracaal94@gmail.com (C.A.S.-R.); nrocha@itdurango.edu.mx (N.E.R.-G.); agallegos@itdurango.edu.mx (J.A.G.-I.); rubenfgl@itdurango.edu.mx (R.F.G.-L.); sgonzalez@itdurango.edu.mx (S.M.G.H.); kherrera@itdurango.edu.mx (K.M.H.-R.); manuel.chairez@itdurango.edu.mx (M.H.C.-R.); 2Facultad de Ciencias de la Cultura Física y Deporte (FCCFyD), Universidad Juárez del Estado de Durango (UJED), Lic. Hector García Calderon S/N. Fracc. SARH, Durango 34113, Mexico; manuel.gonzalez@ujed.mx

**Keywords:** rehydrating beverages, probiotics, agavins, *Q. sideroxyla*, antioxidant, anti-inflammatory

## Abstract

High rehydration beverage consumption represents a significant opportunity for the integration of biotic products that offer the potential to improve body composition and intestinal health. *Quercus sideroxyla* (IQS) infusions contain polyphenolic compounds with antioxidant and anti-inflammatory properties, and in combination with probiotic strains and prebiotic materials, they offer a promising alternative for generating designer beverages for physically active people. These beverages were formulated using a combination of IQS, agave fructooligosaccharides (FOS), microencapsulated probiotics of *Akkermansia muciniphila* and *Bifidobacterium longum*, electrolytes, and glucose. Stable microencapsulated probiotics were obtained by spray drying, using agave gums (PD > 10) and gum arabic as wall materials. The beverage formulations were generated with different percentages of FOS (A:1.6%, B:1.2%, and C:0.8%). The phenolic profile of the beverages was determined by LC-MS/MS, indicating a difference in the concentration of compounds, highlighting changes associated with the addition of FOS compared with IQS. Sensory analyses indicate a preference for the beverage with the highest FOS concentration. The antioxidant potential of the formulations, determined by ABTS, DPPH, and ORAC, showed no differences between the drinks; however, analyses indicate a positive correlation with quinic acid, t-cinnamic acid, quercetin 3-O-glucoside, and total phenolic content, suggesting a synergistic effect. The drinks with higher FOS content exhibited a higher anti-inflammatory potential (EMA). Therefore, it can be concluded that a rehydrating drink with a higher FOS content offers a prebiotic effect with potential anti-inflammatory activity and, according to the panelists, is a suitable drink for evaluating its effects on body composition and intestinal health in people who have recently started physical activity.

## 1. Introduction

The growing interest of the population in recreational physical activity, as well as in the consumption of supplements and beverages, is prompted by the desire to change physical appearance and body composition, especially among previously sedentary individuals. These individuals are exposed to environmental conditions that predispose them to overweight, obesity, inflammation, and oxidative stress, including an inadequate diet and lack of regular physical activity. While physical activity is currently recognized as a primary strategy for body composition modulation, its implementation in individuals with overweight, obesity, or excess adiposity may present certain challenges due to their susceptibility to pre-existing endogenous factors. These include intestinal dysbiosis, characterized by an increased prevalence of pathogenic species with pro-inflammatory effects at the intestinal level, as well as chronic low-grade systemic inflammation associated with excess adiposity [1,2].

The sports and fitness market offers a variety of products designed for athletes and individuals engaged in regular physical activity. These products include rehydration beverages, which are electrolyte-rich fluids intended to restore optimal hydration levels following the loss of fluids and electrolytes caused by physical exertion, exercise, or hot environmental conditions [3]. According to established data, worldwide sales of this type of product are expected to reach USD 112.2 billion by 2025 and to grow annually by 4.95% (Statista, 2024a) [4]. This increase is part of a broader trend in consumer interest in healthy alternatives, as evidenced by the significant growth in the global market value of functional foods and beverages, which reached USD 281,140 million in 2021 (Statista, 2021) [5]. Rehydration beverages are designed to restore the body’s water electrolyte balance more efficiently than water alone due to the presence of glucose and sodium, facilitating water absorption in the small intestine through sodium–glucose transport [6]. These products are particularly beneficial for individuals who engage in physical activity. However, the most popular products on the market often exceed recommended caloric and sugar intake levels, without additional benefits to rehydration [7].

Therefore, it is imperative to implement strategies for the formulation of rehydrating beverages as design products, with a focus on satisfying the nutritional needs of beginner individuals and providing potential health benefits through the integration of biotic products. In this regard, infusions made from *Quercus sideroxyla* leaves have been extensively studied for their antioxidant therapeutic potential in vitro, as well as their anti-inflammatory effect in murine models of colorectal cancer. Promoting the decrease in pro-inflammatory markers such as activated B cell nuclear factor kappa light chain enhancer (NFKβ), tumor necrosis factor alpha (TNFα), interleukin 8 (IL 8), cyclooxygenase 2 (COX 2), and increasing anti-inflammatory markers such as interleukin 10 (IL 10) and cyclooxygenase 1 (COX 1) [8,9]. These effects were associated with the presence of polyphenolic compounds, with chlorogenic acid, quinic acid, and shikimic acid being particularly notable due to their abundance. Additionally, catechin, quercetin, and kaempferol glycosides were identified as significant components [10,11]. Research has shown that *Quercus sideroxyla* infusions can effectively reduce weight gain and body fat percentage in animal models of obesity [12]. In addition, in vitro and in silico methods indicate that polyphenolic compounds, such as ellagic acid hexoside, present in these infusions, promote an inhibitory effect against monoamine oxidase A (MAO-A), an enzyme involved in the degradation of serotonin and noradrenaline. This provides a potential benefit for the maintenance of mental health [11].

A variety of biotic products, such as probiotics and prebiotics, have potentially beneficial effects for beginning athletes. In this sense, supplementation with prebiotics such as fructooligosaccharides (FOS) induces significant changes in microbial composition, increasing the production of short-chain fatty acids (SCFAs). In addition, prebiotic supplementation with FOS, which has been demonstrated to induce significant changes in microbial composition, has been shown to increase the production of SCFAs. Specifically, an increase in acetate, propionate, and butyrate was observed. Butyrate, in particular, has been demonstrated to be important for intestinal function as well as for insulin production, and stimulates muscle growth by promoting protein synthesis [13,14].

Probiotics may have multiple functions. As nutribiotics, they participate in the metabolism of macro and micronutrients, while pharmabiotics are associated with potential therapeutic effects, including the regulation of the intestinal microbiota, insulin sensitivity, and reduced circulating levels of cholesterol, triglycerides, and glucose, among others [15,16]. Nutribiotics include next generation probiotics, including *Akkermansia muciniphila*, a strain present in 3–5% of the gut microbiota of healthy individuals. This nutribiotic is characterized by its ability to adhere to and colonize the intestinal mucosa, promoting the maintenance and integrity of the intestinal barrier [17]. This effect is achieved through the degradation of mucins present in the intestinal mucosa, which are degraded by *A. muciniphila* to produce SCFAs, such as acetate, butyrate, and propionate. It also releases monosaccharides from mucins that serve as food for other commensal bacteria present in the gut [18]. Likewise, this bacterium is of great relevance for physically active individuals due to its ability to reduce body weight gain, glucose intolerance, and fat mass gain, as well as thermogenesis-established in vivo models [19,20]. Conversely, *Bifidobacterium longum* is another very important probiotic that is found in high concentrations in the intestines of infants and adults. This probiotic is characterized by its high production of SCFAs such as lactate, acetate, propionate, and, in higher concentrations, butyrate, producing an effect on the regulation of intestinal inflammation by regulating the immune system and improving the function of the intestinal barrier [21]. Other studies have described that *B. longum* produces butyrate using FOS as a substrate [22]. Butyrate is of special interest because of its effects on insulin resistance and stimulation of protein synthesis, which have been associated with effects such as decreased body fat gain and increased muscle mass gain [23].

The objective of the current study is to develop a rehydrating beverage using *Quercus sideroxyla*, fructooligosaccharides from agave, and microencapsulated probiotics from *Akkermansia muciniphila* and *Bifidobacterium longum*, to establish its antioxidant and anti-inflammatory functional potential.

## 2. Materials and Methods

### 2.1. Chemical Reagents

Phenolic acid standards (quinic, shikimic, gallic, ellagic, protocatechuic, chlorogenic, vanillic, caffeic, syringic, coumaric, ferulic, benzoic, t-cinnamic, 3,4-di-caffeoylquinic, and rosmarinic), flavonoid standards (catechin, epicatechin, procyanidin B1, quercetin 3-O-ß-glucuronide, quercetin 3-O-glucoside, kaempferol 3-O-glucoside, and taxifolin), and 2,4,6-trihydroxybenzaldehyde, fluorescein, 6-hydroxy-2,5,7,8-tetramethylchromane-2-carboxylic acid (TROLOX), 2,20-azobis (2-amidinopropane) dichlorohydrate (AAPH), ammonium persulfate, 2,20-azino-bis(3-ethylbenzothiazoline-6-sulfonic acid) diammonium salt (ABTS), 2,2-diphenyl-1-picrylhydrazyl (DPPH); acetone, methanol, and acetonitrile were LC-MS grade from J.T. Baker (Radnor, PA, USA) Monobasic potassium phosphate, dibasic potassium phosphate, and sodium chloride were purchased from Fermont products.

### 2.2. Raw Material Collection and Processing

The herbal material was identified by the botanist Socorro González-Elizondo from the herbarium of CIIDIR-IPN unidad Durango. *Quercus sideroxyla* Bonpl. leaves were collected at Pueblo Nuevo, Durango (Longitude: 105°21′43″ E Latitude: 23°46′46″ N) from the July 2023 season, with identification number 61,484. The collected plant material was sanitized with 1% sodium hypochlorite for 5 min. The material was then dried in the dark at room temperature (25 °C ± 2 °C), ground to a particle size of 150 µm, and stored until use.

### 2.3. Biological Material

The probiotic strains *Akkermansia muciniphila* (ATCC BAA-835), a Gram-negative bacterium of the Verrucromicobia class, and *Bifidobacterium longum* (ATCC 15707), a Gram-positive bacterium belonging to the Actinomycetes class, were used to produce the rehydration drinks. These strains were obtained from the CONAHCYT National Laboratory for the Support of Biotic Product Evaluation (LaNAEPBi) at the Tecnológico Nacional de México/I.T. de Durango.

### 2.4. Preparation of Quercus sideroxyla Leaves Infusion

The preparation of the infusion from *Quercus sideroxyla* leaves (IQS) was according to Rocha-Guzmán et al. [8]. Briefly, starting from sanitized and dried leaves ground to a particle size of 150 µm, infusion was prepared at 80 °C for 10 min in purified water at a 1% (*w*/*v*) ratio and then filtered. The infusions were frozen at −19 °C and then freeze-dried in a Labconco (Kansas, MO, USA) Freezone 6 plus at 0.046 mBar pressure for subsequent analysis.

### 2.5. Phenolic Profiling by UPLC-ESI-MS/MS of 1% Infusion of Quercus sideroxyla and Rehydration Beverages

The identification and quantification of phenolic profiling of the IQS and rehydrating beverages were performed according to the methodology established by Diaz-Rivas et al. (2018) [24]. Starting from 10 mg of the lyophilized samples, a resuspension was performed in 1 mL of methanol, the samples were vortexed and placed in an ultrasonic bath for 10 min, filtered at 0.45 µm, and deposited in amber vials. The experimental run was performed on an Acquity UPLC system (Waters Corp., Milford, MA, USA) coupled to a Xevo TQ-S triple quadrupole tandem mass spectrometer (Waters Corp.). Data were collected in multiple reaction monitoring (MRM) mode. Data acquisition and processing were performed using MassLinx v.1 software from Waters Corporation. Chromatographic separations were performed on a C18 Acquity UPLC BEH column (100 mm × 2.1 mm × 1.7 µm) (Waters Corp.) operated at 30 °C with a blank temperature of 10 °C, using water/formic acid 7.5 mM (A) and acetonitrile (B) as mobile phases at 250 µL/min and a sample temperature of 5 °C. The gradient was applied in the following manner: at 5% B, an isocratic flow of 0.8 min was maintained, followed by 10% B at 1.2 min, an isocratic fluid of 0.7 min, followed by 15% at 2.4 min, an isocratic fluid of 1.3 min, followed by 21% B at 4.0 min, an isocratic fluid of 1.2 min, followed by 27% B at 5.7 min, 50% B at 8.0 min, 100% B at 9.0 min, returning to initial conditions (5% B) at 11.5 min and maintaining these same conditions until 13.5 min. Negative ionization was used for the MS assay. The ESI conditions consisted of a capillary voltage of 2.5 kV, a desolvation temperature of 300 °C, a temperature source of 150 °C, a desolvation and cone gas of 500 and 151 L/h, respectively, and a collision gas of 0.13 mL/min. MRM transitions were determined by MS/MS spectra of existing phenolic acid standards, and a mixture of different phenolic compounds was used as a monitor of retention time and *m*/*z* values. The identification of the peaks was based on the comparison of their retention times and MRM transitions with those of pure standards. Quantitative determinations of phenolic compounds were executed using standard calibration curves of available standards.

### 2.6. Elaboration and Characterization of Akkermansia muciniphila and Bifidobacterium longum Microcapsules

The pre-inoculum of the prebiotic strains was obtained from selective media, MRS supplemented with 0.05% cysteine for *Bifidobacterium longum*, and anaerobic basal medium supplemented with 0.5% glucose for *Akkermansia muciniphila* at 10% for 24 h at 37 °C under anaerobic conditions. Subsequently, 175 mL of fresh medium containing 10% of the pre-inoculum was added and the mixture was incubated under the same conditions for 12 h. Then, the culture was centrifugated at 2500 rpm for 5 min. The cell pellet obtained was washed with a saline solution (NaCl 8.5%) and then centrifugated at 2500 rpm. This process was repeated three times to eliminate the culture medium. Subsequently, a 10% solution of agavins and gum arabic was added for microencapsulation in a Mini Spray Dryer Buchi 290 spray dryer (BUCHI, Flawil, Switzerland) at a drying temperature of 140 °C with a flow rate of 6 mL/min.

### 2.7. Viability of Microencapsulated Akkermansia muciniphila and Bifidobacterium longum

Aliquots of 10 mg were taken and resuspended in 1 mL of peptone water. Serial dilutions of 10^−6^, 10^−7^, and 10^−8^ were made, inoculated into the selective media, and incubated under anaerobic conditions for 18 h at 37 °C. Survival was determined according to the following formula:$$Survival=\frac{\left(viable\;count\;per\;gram\;of\;microcapsules)(total\;grams\right)}{initial\;viable\;count\;(in\;100\;mL\;of\;suspension)}\times 100$$

### 2.8. Physicochemical Quality of Symbiotic Microcapsules

#### 2.8.1. Wettability

A sample (1 g) of microencapsulated probiotics was dispersed in a beaker containing 450 mL of water at room temperature. The time required for the sample to be completely absorbed into the liquid was registered.

#### 2.8.2. Water Activity (aw)

A sample (3 g) of probiotic microcapsules were taken and placed in a Hygrolab AW-DIO water activity analyzer (ROTRONIC International, Hauppauge, NY, USA).

#### 2.8.3. Moisture

A sample (2 g) of microcapsules were weighed on an OHAUS Mb 23 automatic moisture analyzer (Parsippany, NJ, USA).

#### 2.8.4. Solubility

A sample (1 g) of microcapsules were dissolved in 100 mL of distilled water, shaken with a blender at maximum speed for 1 min, centrifuged at 3000 rpm for 5 min. Aliquots of 25 mL were taken and placed in Petri dishes at constant weight for 5 h at 105 °C, and the final weight was registered.

### 2.9. Survival of Microencapsulated Akkermansia muciniphila and Bifidobacterium longum to Oro-Gastrointestinal Digestion

The in vitro digestion process was performed according to established methodology [25]. A sample (1 g) of the microencapsulated probiotics was dissolved in 15 mL of gastrointestinal electrolyte solution GES (NaCl, KCl, NaHCO_3_, and CaCl_2_). The solution was stirred manually for 2 min. In the oral phase, 300 µL of lysozyme was added and shaken manually for 2 min. To simulate the gastric phase, 375 µL of pepsin was added, and the pH was adjusted to 3.0 with 1 M HCl. The mixture was stirred for 30 min at 37 °C and 180 rpm, then the pH was adjusted to 2.0. In the intestinal phase, the pH was adjusted to 6.5–6.7 with NaHCO_3_ 1 M, 350 µL of bile salts, and 875 µL of pancreatin, and the mixture was stirred for 60 min under the same conditions. Finally, 10 mL of IES (NaCl, KCl y CaCl_2_) was added to the 100 mL aliquot, and the mixture was shaken for 60 min. At each stage, an aliquot was taken for serial dilution and subsequent plate seeding by surface extension in the selective media for each prebiotic.

### 2.10. Formulation of Functional Rehydrating Beverages from Infusions of Quercus sideroxyla Leaves, Prebiotics, and Microencapsulated Probiotics

Three rehydrating beverages were formulated from the infusion of 1% *Q. sideroxyla* leaves, microencapsulated probiotics added with electrolytes and different concentrations of glucose and fructooligosaccharides (FOS) at a final volume of 500 mL (Table 1).

### 2.11. Proximate and Physicochemical Analysis of Rehydrating Beverages

A proximate analysis was performed using the AOAC (2005) [26] methods by means of the determinations of moisture (AOAC 964.22), ash (AOAC 962.09), protein (AOAC 984.13), and carbohydrate by difference. The pH was evaluated according to the official Mexican standard NMX-F-317-NORMEX-2013 [27] and the total soluble solid content (°Brix) according to the Mexican standard NMX-F-112-NORMEX-2010 [28].

### 2.12. Sensory Analysis of Rehydrating Beverages

A sensory analysis was performed with the participation of 29 individuals after evaluation and approval by the research ethics committee of the Tecnológico Nacional de México under project code CEI-003-2022-0301-023. The sensory evaluation was divided into two stages. A sample corresponding to 20 mL of formulations A, B, and C was provided for analysis by the panelists using the following tests:(a)Free choice of profile: From the most concentrated sample of agavins (10 g/dose), the main attributes to be evaluated were identified and generated. For this purpose, 20 untrained panelists participated, administering a volume of 20 mL of each sample.(b)Sorting test: samples were sorted according to acceptability. It was carried out with a total of 20 untrained panelists, administering a volume of 20 mL of each sample.(c)Focus groups: this test was used to measure the acceptability of each rehydrating drink formulation, using a 9-point hedonic scale where 1 = I dislike it very much and 9 = I like it very much. For this study, 20 untrained panelists participated, administering a volume of 20 mL of each sample.(d)Quantitative descriptive analysis: the magnitude of the attributes identified in the free choice profile test was evaluated by means of a nine-point intensity scale where 1 = not at all and 9 = very intense, 20 untrained panelists participated, administering a volume of 20 mL of each sample.

### 2.13. Total Phenolic Content in Rehydrating Beverages

The determination of total phenolics was performed by the Folin–Ciocalteu [29]. Briefly, 25 µL of the beverages, 80 µL of distilled water, and 5 µL of Folin’s reagent were added in a 96-well plate and subsequently incubated in dark conditions at room temperature for 5 min. An aliquot (80 µL) of Na_2_CO_3_ solution was added, shaken, and allowed to stand 30 min in dark at room temperature to subsequently perform the readings at 750 nm in a Synergy HT microplate reader (Bio-Tek, Winooski, VT, USA).

### 2.14. Total Flavonoid Content in Rehydrating Beverages

The determination of total flavonols was performed by the aluminum chloride reaction method according to the methodology described by Muñoz et al. (2014) [30]. Briefly, 20 µL of the beverages, 6 µL of NaNO_2_, 12 µL of AlCl_3_, 40 µL of NaOH, and 122 µL of distilled water were added in a microplate, incubation was performed in dark conditions at room temperature for 5 min to subsequently perform readings at 510 nm in a Synergy HT microplate reader (Bio-Tek, Winooski, VT, USA).

### 2.15. Evaluation of the Antioxidant Potential of Functional Rehydrating Beverages

#### 2.15.1. Determination of Antioxidant Capacity by the ABTS Method

The determination of the ABTS assay was performed accordingly [31]. In a 96-well plate, 10 µL of sample, curve or blank, was placed, 190 µL of ABTS radical was added and incubated at room temperature for 10 min in the dark. After incubation, the reading was performed at 750 nm in a Synergy HT microplate reader (Bio-Tek, Winooski, VT, USA).

#### 2.15.2. Determination of Antioxidant Capacity by the DPPH Method

Antioxidant potential was determined by the DPPH assay accordingly [32]. DPPH radical was prepared immediately prior to use by being dissolved at a concentration of 100 µM in 100% methanol, always protecting it from light. An aliquot (50 µL) was placed in a microplate, 150 µL of DPPH was added and gently shaken and protected from light and read after 30 min incubation at an absorbance of 515 nm on the Synergy HT microplate reader (Bio-Tek, Winooski, VT, USA).

#### 2.15.3. Determination of Antioxidant Capacity by the ORAC Method

The ORAC assay was performed accordingly [33]. Briefly, 20 µL of sample was placed in a dark 96-well plate, 200 µL of fluorescein solution (14 μM) was added to the sample, incubated for 15 min at 37 °C, and 75 µL of AAPH solution was added and a 3 h kinetic was performed, measuring every 3 min at 480 nm excitation and 580 nm emission in a Synergy HT microplate reader (Bio-Tek, Winooski, VT, USA).

### 2.16. Determination of Anti-Inflammatory Capacity by Erythrocyte Membrane Stabilization (EMA)

The anti-inflammatory activity was determined using established methodology [34]. First, a 10% suspension of erythrocytes in saline (0.85%) was prepared from fresh blood obtained from a healthy human donor with informed consent. A test mixture was then prepared containing 200 µL of hypotonic solution (0.36% *w*/*v* NaCl), 100 µL of phosphate buffered saline (PBS, 10 mM, pH 7.4), 100 µL of the rehydration beverage, and 50 µL of 10% red cell suspension. Two controls were included: one replacing the extract with normal saline and the other replacing the erythrocyte suspension with normal saline. Indomethacin at 2.5 mg/mL was used as the standard drug. The reaction mixtures were incubated at 56 °C for 30 min under agitation followed by centrifugation at 5000 rpm for 10 min at 4 °C. The supernatants were recovered, and their absorbance was measured at 560 nm using a Synergy HT plate reader (Bio-Tek, Winooski, VT, USA). The calculation of the percentage of membrane stabilization was performed using the following formula:% membrane stabilization = 100 − (Abs sample − (Abs control 1)/(Abs control 2)) × 100

### 2.17. Statistical Analysis

Data were expressed as mean ± standard deviation. For normal trend data, a one-way ANOVA analysis of variance was used considering the comparison of means according to Tukey’s test (*p* < 0.05). Data with a trend different from normal were analyzed according to the Mann–Whitney U-test (*p* < 0.05). Multivariate analysis was performed by partial least squares discriminant analysis (PLS-DA) as well as Spearman correlation ranking analysis, considering a correlation score equal to or greater than 0.6 as strongly correlated using Metaboanalyst 6.0 software.

## 3. Results

### 3.1. Phenolic Profiling of Quercus sideroxyla Leaf Infusion

The basis of the formulation of the rehydration beverages is a 1% infusion of *Quercus sideroxyla* (IQS) leaves. It has been reported that these infusions generate beneficial effects related to the reduction of oxidative and anti-inflammatory processes due to the content of polyphenolic compounds. In this sense, the profiling of polyphenolic compounds was monitored, analyzing 29 phenolic acids, 21 flavonoids, and 12 hydrolysable tannins. Of these, 16 phenolic acids, 16 flavonoids, and 9 hydrolysable tannins were identified (Table 2). The phenolic profiling shown by IQS is like that shown by previous authors, highlighting in the profiling obtained a higher abundance of quercetin glucuronide [11]. It is important to consider that factors such as storage time, seasonality, and region of production or hot weather can modify the concentration of polyphenolic compounds [35].

In terms of abundance, phenolic acids such as chlorogenic acid, quinic acid, and 4-O-caffeoylquinic acid were identified. Chlorogenic acid has been described as a potential regulator of adipose tissue gain via modification of fatty acid and triglyceride metabolism [36].

Quercetin and its derivatives, including quercetin glucuronide and quercetin 3-O-glucoside, have been identified as key compounds in this formulation. These compounds have previously demonstrated efficacy in improving clinical conditions, such as insulin resistance, by activating the AMPK pathway, which plays a crucial role in protein synthesis [37]. Additionally, they have been characterized for their antioxidant activity in vivo, stimulating the production of endogenous antioxidants such as superoxide dismutase (SOD) and catalase (CAT), as well as anti-inflammatory activity by modulating tumor necrosis factor alpha (TNFα), interleukin 1 beta (IL1β), and cyclooxygenase 2 (COX2) [38,39,40].

### 3.2. Characterization of Akkermansia muciniphila and Bifidobacterium longum Microcapsules

To evaluate the quality of the microencapsulation process of *Akkermansia muciniphila* and *Bifidobacterium longum* to obtain symbiotic microcapsules, the spray-drying yield, viability, and survival of the probiotics were established, as well as the physicochemical determinations of the microcapsules (Table 3). The spray drying of both strains using the mixture of agavins and gum arabic presented yields above 62.86% for *A. muciniphila* and 62.25% for *B. longum*, which can be considered as adequate values, considering that the microencapsulation of probiotics by this operation usually produces low yields around 43% [41]. Regarding the viability of the study strains, yields higher than 10.40 log cycles per gram of microencapsulate were obtained, representing a survival higher than 97% for both strains. These levels are similar to those obtained by other authors who reported 96% survival of *B. breve* and *L. lactis* by spray drying using 15% agavins as wall material at 150 °C as initial temperature [25].

Also, other researchers have reported high survival rates to the spray drying in species of the genus *Bifidobacterium*, which demonstrate an intrinsic resistance to exposure to oxygen and heat during the process, as the high survival of *B. longum* obtained [42]. Likewise, it is important to consider that the inclusion of gum arabic as a wall material for microencapsulation has been shown to protect the core of the microcapsules during the spray-drying process, presenting greater protection against heat, as well as greater encapsulation efficiency derived from its structure [43].

Additionally, the physicochemical parameters of microencapsulates were assessed, yielding water activity values of 0.118 for *A. muciniphila* and 0.117 for *B. longum*, and moisture contents of 3.15% for *A. muciniphila* and 2.85% for *B. longum*. These values are within the adequate ranges described for this type of powder with water activity values below 0.6 and humidity levels below 5%. This indicates that the process was highly effective in enhancing its stability during storage, mainly by preventing fungal contamination and enzymatic degradation [44]. Additionally, stability studies were performed on the microencapsulated probiotics for 90 days at room temperature. The results indicated that there were no statistically significant changes in viability, stability, or physicochemical parameters in both strains (Appendix A). However, it would be appropriate to extend this stability study for more than 12 months.

### 3.3. Gastrointestinal Survival of Akkermansia muciniphila and Bifidobacterium longum Microcapsules

It has been established that in vitro digestion studies are an alternative to establishing the possible structural changes of microencapsulated probiotics during the digestion process, so that it is possible to predict the viability of probiotics upon reaching the colon and predict their beneficial effect [45]. After performing the simulated digestion process, it was determined that at the oral level, the microencapsulated probiotics presented a decrease of 0.01 logarithmic cycles for *A. muciniphila* and 0.05 for *B. longum*, while the free cells were 0.11 and 0.08 for both strains, respectively.

At the gastric level, microencapsulated cells did not show a significant decrease compared to free cells of 2.68 logarithmic cycles for *B. longum* and 2.49 logarithmic cycles for *A. muciniphila*. This indicates that microencapsulation with agavins and gum arabic promoted the protection of probiotics in conditions of acidity and gastric motility, decreasing only 1.22 and 1.03 logarithmic cycles, respectively (Table 4).

Therefore, it can be suggested that the microencapsulation process maintains the viability of probiotic strains of both species when they reach the colon. For the *A. muciniphila* strain, the intestinal survival was over 87%, and for *B. longum,* over 86%, higher than the 65% shown by both strains subjected to digestion without prior microencapsulation. These results are superior to those shown by others, who reported a survival of about 66% for the *L. casei* strain and 81% for the *B. breve* species [25]. Therefore, the microencapsulation process of both strains manages to protect them from the digestive process.

In this sense, it is expected that when probiotics reach the colon, beneficial effects are generated in the host organism. These effects are mediated by the production of bacterial metabolites such as SCFA, deconjugation of secondary bile acids, and even improvement of mineral absorption [46]. In the specific regard of *Akkermansia muciniphila*, it has been characterized by improving the intestinal barrier through the production of mucin and the stimulation of mucin production in goblet cells in the colon, as well as the effect of reducing pathogenic bacteria [47]. In addition, *Bifidobacterium longum* has been associated with effects such as colonizing the large intestine, promoting intestinal integrity by stimulating the production of tight junction proteins (TJPs), and producing butyrate and acetate in the presence of fructooligosaccharides [48].

### 3.4. Nutritional Characteristics of Rehydration Beverages

As part of the characterization of rehydration beverages, a proximal analysis was accomplished, as well as physicochemical determinations that contributed to establishing the nutritional characteristics of the beverages (Table 5). The results indicate that the formulations of the three drinks provide a lower amount of carbohydrates (2 g) per 100 mL of product compared to the widely consumed products on the market, Gatorade and Powerade, which provide 4 and 4.2 g/100 mL of product, respectively. Consequently, the total carbohydrate content per 100 mL is 1.6, 1.2, and 0.8 g, corresponding to agave fructooligosaccharides, which are dietary fiber and not sugars, in comparison with the consistent sugar content of 100% of the most widely consumed beverages on the market. In comparison with the commercial leading drinks, the formulations developed have a more suitable nutritional profile for individuals with lower carbohydrate demands, such as people who have recently started to do physical activity. However, it is possible to modify this formulation to adapt it to the needs of high-performance athletes, not only in terms of electrolyte content, but also in terms of carbohydrate content for glycogen replacement.

### 3.5. Acceptability and Sensory Characteristics of Rehydrating Beverages

To establish the effect of the formulation of rehydration beverages on sensory acceptability, several tests were followed up. In this case, the main factor of variation was the content of agavins as a source of carbohydrates in the beverages. In the free profile test, which is the preferred method for assessing attributes perceived by untrained panelists, it was determined that formulation A, enriched with high concentrations of agavins, led to a decrease in attributes associated with a salty taste. In contrast, formulation C, with its lower carbohydrate content and higher dextrose proportion, resulted in an increased frequency of attributes perceived as sour and herbal (Table 6). Quantitative descriptive analysis measured attributes on a scale of perceived intensity from 0 to 9, establishing the refreshing sensation and sour taste as the main attributes of the beverage, while the sweetness attribute was perceived as low in all formulations. These results could be related to the low glucose content, as well as to the low sweetening power of fructooligosaccharides (FOS), which is less than 50% with respect to sucrose [49].

In addition, the different attributes can be related to the effects of polyphenolic compounds present in the infusions. Specifically, a principal component analysis was carried out, determining that the score graph shows a similar behavior between formulations B and C with respect to the perceived attributes, unlike formulation A (Figure 1a). In addition to the analysis of the reduction of dimensions of the variables, it was possible to establish various correlations of the polyphenolic compounds with the sensory attributes of the drink (Figure 1b). A correlation was determined between the total phenolic content and the presence of phenolic acids such as t-cinnamic and hydroxybenzoic acids with the grassy flavor. Additionally, a positive correlation was determined between FOS and a heightened perception of sweet flavor attributes, along with a refreshing sensation. However, a masking or negative relationship was observed with salty flavors. This finding relates to other research using FOS in beverages, where it was shown that adding lemon enhances the perception of sweetness. This could be attributed to the increased sweetness in formulation A, which contains a higher proportion of FOS, as was also reported in rehydration drinks with citric acid exhibiting a similar effect [50].

Conversely, a correlation has been established between bitter taste and chlorogenic acid and quercetin glucuronide. Similar results have been reported, indicating that phenolic acids, such as p-coumaric acid, are associated with the presence of this attribute [51].

On the other hand, as part of the acceptability tests for rehydration drinks, the acceptability of the drinks was established through a study of ordering and general acceptability in directed groups. In this study, it was established that there were no differences between the three formulations developed. The beverages demonstrated adequate acceptability according to the six points of the scale. However, the ordering test indicates that formulation A is the most acceptable, since it achieves a higher parametric score compared to formulations B and C (Figure 2). It is important to emphasize that rehydrating beverages often have low acceptability due to salty and sweet flavors [52].

### 3.6. Evaluation of the Phenolic Content and Profiling of the Rehydrating Beverages

To evaluate the effect of the components added to the rehydrating beverage formulations on the total content of phenolic compounds and flavonoids, both determinations were monitored (Figure 3). The results indicate that there are no statistically significant differences in total phenolic and flavonoid contents between the rehydrating beverages. The beverages had total phenolic concentrations of 8.81 ± 0.10, 8.29 ± 0.09, and 8.22 ± 0.26 mg/mL, respectively, which are lower than the IQS concentration of 12.46 ± 0.10 mg/mL.

Likewise, the concentrations of total flavonoids were similar among the beverages (4.61 ± 0.24, 4.47 ± 0.23, and 4.47 ± 0.33 mg eq. gallic acid/mL) but lower compared to IQS (5.23 ± 0.10 mg eq. gallic acid/mL). This result may be related to the addition of agave fructooligosaccharides as part of the formulation, since the interaction of the hydroxyl functional groups present in the fructans, as well as their branched structure, allows them to interact with the phenolic compounds of the IQS that is part of the beverage, generating a complexation [53]. Previous reports have established the complexation of polyphenols with polymers such as starch, demonstrating that this effect protects polyphenol compounds from degradation during the digestion process [54]. 

As part of the monitoring of the polyphenolic compounds present in the rehydrating beverages, a lower presence of these compounds was determined compared to those identified in IQS, identifying 11 phenolic acids and 10 flavonoids (Table 7). These results correlate with the detection of lower levels of total phenolic acids and flavonoids. In the formulations, quinic and chlorogenic acids are abundant. Chlorogenic acid has been identified for its ability to significantly increase the activity of antioxidant enzymes such as superoxide dismutase (SOD) and catalase (CAT) [55]. In addition, this acid has been characterized by a high antioxidant capacity in vitro, since it promotes the neutralization of free radicals such as superoxide, hydroxyl, and peroxyl [56]. In vitro studies have determined that quinic acid can inhibit lipid peroxidation in cell membranes, thus contributing to protection against oxidative stress [57]. This compound has also been associated with the stimulation of endogenous antioxidant mechanisms, derived from its molecular structure which contains multiple hydroxyl groups capable of donating electrons to reactive oxygen species (ROS) [58].

The most abundant flavonoids identified in the beverages were catechin and quercetin glucuronide. These compounds have been characterized by their antioxidant and anti-inflammatory activities. Specifically, catechin has been noted for its anti-inflammatory capacity by inhibiting cyclooxygenase 2 (COX 2) and lipoxygenase (LOX), which reduces the synthesis of inflammatory mediators such as prostaglandins and leukotrienes, in addition to its antioxidant activity [59,60].

In contrast, it was determined that the rehydrating beverages with higher concentrations of fructooligosaccharides (FOS) (A and B) had significantly lower concentrations of 4-O-caffeoylquinic, gallic, and shikimic acids. Similarly, flavonoids exhibited a similar reduction, primarily in the aglycone forms of quercetin, naringin, and phloridzin, with only traces of these compounds being found in the rehydrating beverages. It is important to consider that complexation reactions have been reported between fructans and polyphenols, which depend on their structure and affinity for fructans. Although this affects the accessibility of polyphenols [61], the complexation may decrease the functional potential in vitro. However, in vivo studies have been associated with the protection of compounds during gastrointestinal transit [62]. This complexation has a positive effect on the fermentation process followed through by the intestinal microbiota, since the latter has enzymes that allow the release of polyphenols from the aglycones and their subsequent biotransformation into metabolites with greater bioaccessibility and bioavailability to promote diverse biological effects, including antioxidant and anti-inflammatory effects [62]. Prebiotic effects of polyphenolic compounds have also been described, as they serve as a substrate for bacteria with positive effects on body composition and intestinal health, such as *Akkermansia muciniphila*, *Bifidobacterium*, *Lactobacillus*, among other strains [63,64,65].

### 3.7. Antioxidant Potential of Rehydration Beverages

The antioxidant potential of beverages is crucial for people starting to engage in physical activity as respiratory flow and oxygen demands increase [66]. The results of antioxidant tests of the three formulations were similar in terms of trapping different radicals (Figure 4a). The greatest antioxidant power shown by the three formulations is observed in the ORAC and DPPH assays, the former being mainly associated with the ability of phenolic compounds to delocalize electrons. This effect could be associated with the presence of compounds such as quinic acid, trans-cinnamic acid, and quercetin 3-O-glucoside, which have been described as having a high antioxidant effect [67]. Polyphenolic compounds exhibit antioxidant activity by suppressing the generation of free radicals, thereby effectively reducing oxidation rates through two mechanisms: the inhibition of the formation of active species and the deactivation of free radical precursors [68].

However, when comparing the antioxidant effect of the three formulations, they have a lower effect in relation to the IQS. These results may be related to a lower polyphenolic content determined in phenolic acids, flavonoids, and hydrolysable tannins, due to the concentration of IQS used for the generation of the rehydrating drink, which was only 50 percent of the total volume. It is also important to highlight that the interaction between FOS with polyphenol compounds in the formulated drinks affects their capacity to delocalize electrons, i.e., to accept H ions from the OH groups present, therefore affecting their functional potential as an antioxidant in vitro [69]. However, as previously described in in vivo models, this complexation favors the protection of polyphenol compounds, allowing them to reach the colon and be biotransformed by the microbiota, being able to produce an antioxidant effect. This can be attributed to either the inhibition of free radicals or the induction of antioxidant pathways, such as erythroid 2 (NRF2), which is related to the expression of antioxidant enzymes [70].

### 3.8. Anti-Inflammatory Potential of Rehydration Beverages

In people who have recently engaged in regular physical activity, and usually starting from a pre-existing sedentary lifestyle, there is a frequent tendency towards systemic inflammation. This has been linked to changes in body composition, such as an increase in the proportion of body fat, as well as insulin resistance [71]. The stabilization of erythrocyte membranes was evaluated as an indirect measure of anti-inflammatory activity. The assay is based on the principle that inflammation involves the lysis of lysosomal membranes, which leads to the release of inflammatory mediators. Therefore, the stabilization of the lysosomal membrane limits the inflammatory process [72].

As demonstrated in Figure 4b, the three rehydration beverages demonstrate a higher anti-inflammatory potential than the IQS (41.12 ± 0.74) and significantly higher than the anti-inflammatory drug indomethacin. Formulation A, containing a higher percentage of agave FOS (1.6%), showed a significantly higher anti-inflammatory potential (72.64 ± 0.11) compared to formulations B (8.34 ± 0.60) and C (69.14 ± 1.24). Polyphenol compounds from the IQS have been linked to anti-inflammatory effects, including the ability to stabilize biological membranes [73]. Among them are flavonoids such as quercetin, kaempferol, and catechin [74]. Studies on in vitro models have described that polyphenol compounds have the capacity to interact with the non-polar nucleus of the erythrocyte cell membrane, providing a decrease in oxidative stress as well as a decrease in lipid peroxidation, promoting the protection of the erythrocyte membrane [35,74].

Additionally, when FOS is added to rehydration beverages, it has a beneficial effect on the stability of erythrocyte membranes, primarily through its ability to retain water, increasing the osmolarity of the solution and reducing the effect of hypotonicity. Likewise, the laminar configuration of FOS, with its β (2-1) and β (2-6) fructosyl bonds, also contributes to membrane stability [75].

Therefore, based on the results obtained, it is hypothesized that the consumption of the rehydration drink can generate an anti-inflammatory effect due to either its content of phenolic compounds that can inhibit the pro-inflammatory NFKβ pathway, or the effect that the microbial metabolites produced by the FOS added to the beverage can generate. In addition, the microencapsulated probiotics contained in the beverage may contribute to the balance of the intestinal microbiota as well as to the maintenance of the intestinal barrier, which are important components for the generation of pro-inflammatory processes [76].

### 3.9. Stability Study of the Rehydration Drink with the Highest FOS Content

Having established that formulation A had the greatest anti-inflammatory potential and that there were no significant differences in terms of overall acceptability and antioxidant potential, it was decided to conduct a storage stability for a later use in intervention models with physically active people. The results showed that there was a decrease in phenolic content and total flavonoids after 14 days of refrigerated storage, which has a direct impact on the antioxidant potential of the beverages. Similarly, the anti-inflammatory potential of EMAs decreased after 5 days, being related to the variation in pH, which decreases during the same period. Finally, the viable count of *B. longum* and *A. muciniphila* strains was reduced after 14 days of storage, which could be related to a possible release of the microcapsules during storage and their exposure to adverse conditions, especially the presence of hydroelectrolytic salts (Table 8). Other studies have used gum arabic encapsulation as a wall material for microencapsulation, promoting the stability of microcapsules, as it has a pH tolerance in the range of 2 to 8 [77]. On the other hand, the use of gums in spray-drying microencapsulation is often used to maintain stability in media with different pH values with optimal stability values found in the pH range of 4 to 6 [78]. Therefore, it would be recommended to have storage times from 1 to 15 days under refrigerated conditions, so that the potential effects of the drink ingredients are not compromised.

### 3.10. Relationship Between the Composition of Rehydration Beverages and Their Biological Antioxidant and Anti-Inflammatory Potential

To establish a relationship between the phenolic composition and the potential antioxidant and anti-inflammatory effects of rehydration beverage, a partial least squares discriminant analysis (PLS-DA) was carried out to observe the difference in comportment between groups. The analysis indicated that the three rehydration beverages exhibited distinct overall behavior based on their different formulations. In these, formulation B has an intermediate response compared to formulations A and C, which is directly related to the intermediate concentration of agave FOS (1.2%) compared to the 1.6% and 0.8% contained in formulations A and C, respectively (Figure 5a). Additionally, a correlation analysis was accomplished on the composition variables of the rehydration beverage with the antioxidant potential by the ABTS, DPPH, and ORAC assays, as well as the anti-inflammatory potential by means of the evaluation of the stabilization of erythrocyte membranes (EMAs). The analysis was performed using a Spearman correlation study with a correlation value of 0.6 or higher considered as relevant. It was determined that, with respect to the antioxidant potential of the ABTS and ORAC assays, the total phenolic content generates a synergistic effect to promote this effect. Specifically, the results obtained from the ABTS assay correlate directly with the total quinic acid content, while the ORAC assay correlates directly with the t-cinnamic acid content (Figure 5b–d). In this sense, quinic acid has been characterized by its antioxidant capacity due to the presence of OH substituents as well as its side chains [79]. Meanwhile, t-cinnamic acid has shown in previous studies to have an important antioxidant capacity for electron scavenging due to its phenolic ring and hydroxyl substituents [80]. Finally, the antioxidant potential obtained from the DPPH test showed that the total flavonoid content, and the contents of quercetin 3-O-glucoside and quinic acid, were the main variables related to this effect. Other research has described that flavonoids have antioxidant capacities, not only due to the presence of hydroxyl groups, but also due to the ability to delocalize electrons along their three rings, which allow them to enter into resonance, stabilizing reactive species [81].

Finally, only agave fructooligosaccharides (FOS) was the variable showing a strong correlation with anti-inflammatory potential (Figure 5e). In this regard, the main mechanism by which agave fructooligosaccharides could contribute to the regulation of membrane stability, in addition to binding to membrane proteins, is the regulation of the osmolarity of the medium through the presence of hydroxyl functional groups that allow it to interact in solution with water [82].

## 4. Conclusions

The focus of this work was the development of a sensory acceptable rehydration beverage using biotic products. The different formulations developed from the leaf infusions of *Quercus sideroxyla*, microencapsulated probiotics of *Akkermansia muciniphila* and *Bifidobacterium longum* strains, and agave fructooligosaccharides show better values of sensory acceptability and antioxidant and anti-inflammatory potential, which are related to their polyphenolic profile rich in quercetin glucuronide, catechin, and chlorogenic acid. The drink with the highest FOS content was preferred by the panelists and also had the highest anti-inflammatory potential. Therefore, the consumption of this beverage by physically active individuals promises to have an effect on body composition and improve athletic performance through various mechanisms such as the production of postbiotics such as short-chain fatty acids, the modification of the composition of the intestinal microbiota, and the production of anti-inflammatory and antioxidant effects.

## Figures and Tables

**Figure 1 foods-14-00837-f001:**
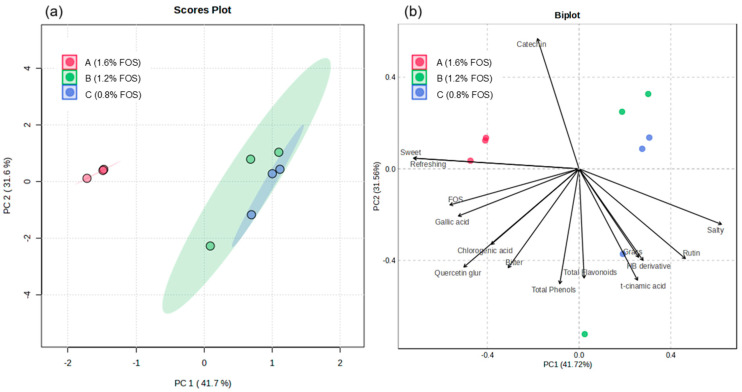
Principal component analysis (PCA) relating the polyphenolic components of rehydration drinks to the sensory attributes perceived by the panelists. (**a**) Scores graph showing the overall behavior of the groups in the model, and (**b**) biplot reducing the dimensions for the relationship of dimensions of the variables.

**Figure 2 foods-14-00837-f002:**
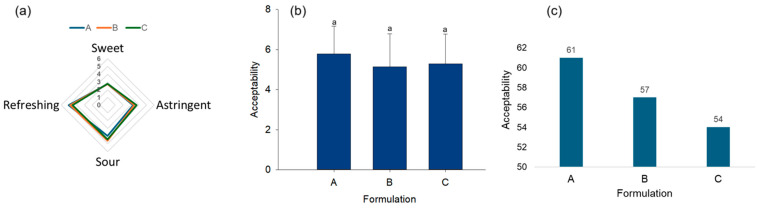
Focus group study of the ordering and general acceptability of the rehydration beverage formulated from IQS, probiotics, and prebiotics. (**a**) Attribute presence scale using quantitative descriptive analysis. (**b**) General acceptability through the analysis of directed groups considering a hedonic scale of 1 to 9. Values expressed as mean ± deviation. (**c**) Levels of acceptability, cumulative score of the ordering test obtained considering the parametric scores of places, place 1: 3 points, place 2: 2 points, and place 3: 1 point. Statistical analysis by Mann–Whitney U-test. A (1.6% agavins), B (1.2% agavins), and C (0.8% agavins).

**Figure 3 foods-14-00837-f003:**
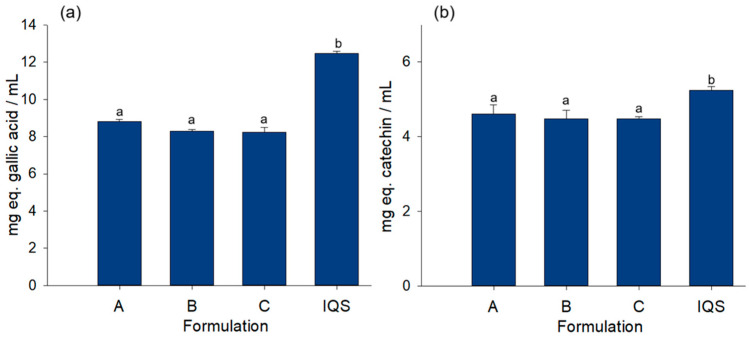
Phenolic content of rehydration beverages formulated from IQS, probiotics, and prebiotics. (**a**) Total phenols and (**b**) Total flavonoids. A (1.6% agavins), B (1.2% agavins), and C (0.8% agavins). Significant differences are expressed by different literals according to Tukey (*p* < 0.05).

**Figure 4 foods-14-00837-f004:**
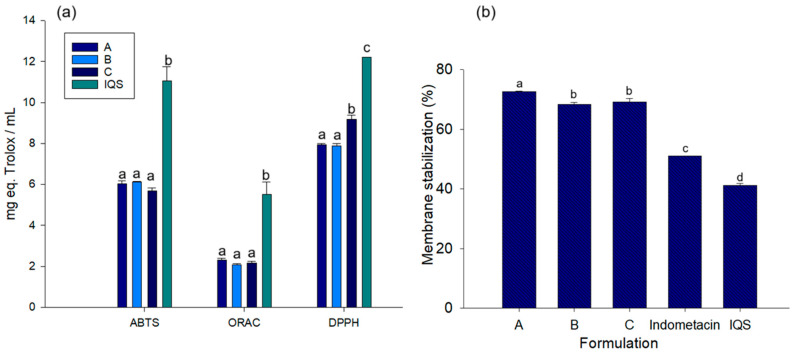
Functional potential of rehydration beverages formulated from IQS, probiotics, and prebiotics. (**a**) Antioxidant potential evaluated by means of ABTS, ORAC, and DPPH assays. (**b**) Anti-inflammatory potential studied by means of a membrane stabilization activity test (EMA). Significant differences are expressed by different literals according to Tukey (*p* < 0.05).

**Figure 5 foods-14-00837-f005:**
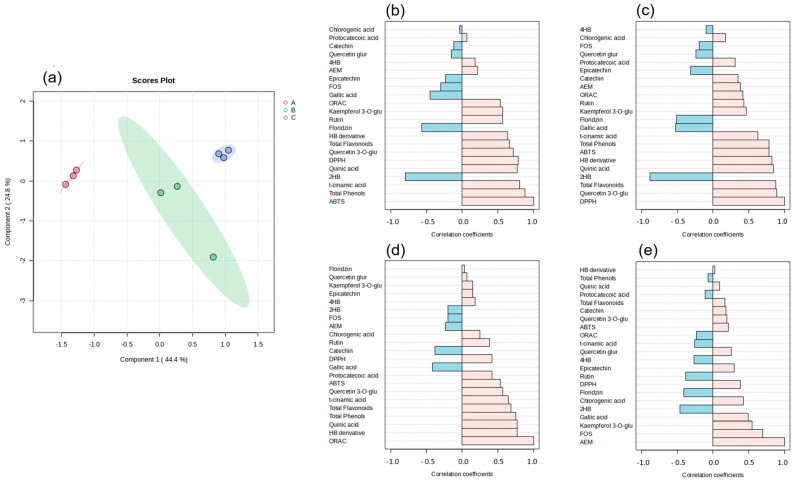
Contribution of the variables in relation to the overall behavior of rehydration beverage. (**a**) Partial least squares discriminant analysis (PLS-DA) score graph. (**b**) Antioxidant potential by the ABTS assay. (**c**) Antioxidant potential by the DPPH assay. (**d**) Antioxidant potential by the ORAC assay, (**e**) anti-inflammatory potential by EMA. Statistical analysis of the Spearman correlation.

**Table 1 foods-14-00837-t001:** Experimental design of the formulation of rehydration beverages.

Ingredients	Formulation		
	A	B	C
*Quercus sideroxyla* infusion 1%	250 mL	250 mL	250 mL
Water	250 mL	250 mL	250 mL
Glucose	2 g	4 g	6 g
Agave fructooligosaccharides (FOS)	8 g	6 g	4 g
Sodium chloride	500 mg	500 mg	500 mg
Potassium chloride	100 mg	100 mg	100 mg
Magnesium chloride	45 mg	45 mg	45 mg
Sodium citrate	75 mg	75 mg	75 mg
Citric acid	65 mg	65 mg	65 mg
Microcapsules *Akkermansia muciniphila* (1.62 × 10^8^ cells)	5.94 mg (3.24 × 10^5^ cells/mL)		
Microcapsules *Bifidobacterium longum* (1.62 × 10^8^ cells)	6.39 mg (3.24 × 10^5^ cells/mL)		

**Table 2 foods-14-00837-t002:** Phenolic characterization of infusion of *Q. sideroxyla* leaves determined by UPLC-ESI^-^-MS/MS.

	Compound	Retention Time (min)	Main Transitions	Concentration (µg/mL)
**Phenolic acid**	4-Hydroxybenzoic acid	3.26	93.05	0.17 ± 0.00
	2,4,6 trihydroxybenzaldehyde	5.76	182.05	0.06 ± 0.00
	2,5-di-hydroxybenzoic acid	2.90	108.92	0.20 ± 0.03
	Syringic acid	4.86	187.21	0.04 ± 0.00
	Protocatechuic acid	2.15	109.05	2.64 ± 0.02
	Vanillic acid	4.54	152.02	0.02 ± 0.00
	Gallic acid	1.11	125.05	28.54 ± 0.02
	Shikimic acid	0.63	111.07	54.92 ± 0.02
	t-cinnamic acid	8.48	103.08	0.04 ± 0.00
	Coumaric acid	4.99	119.08	0.30 ± 0.00
	Ferulic acid	5.63	134.04	0.02 ± 0.00
	Quinic acid	0.57	85.06	169.78 ± 0.04
	Caffeic acid	3.88	135.08	0.22 ± 0.00
	Chlorogenic acid	3.46	191.20	66.04 ± 0.25
	4-O-caffeoylquinic acid	4.45	191.20	32.28 ± 0.01
	Caffeoylquinic acid	2.62	191.20	0.01 ± 0.00326
**Flavonoids**	Naringin	6.90	150.92	0.11 ± 0.01
	Naringenin	9.81	119.00	0.00 ± 0.00
	Taxifolin	6.6	285.00	0.01 ± 0.00
	Phloretin	10.00	176.00	0.30 ± 0.01
	Floridzin	7.39	272.98	0.11 ± 0.00
	Kaempferol	10.26	151.00	0.18 ± 0.00
	Kaempferol glucoside	7.28	284.00	0.36 ± 0.00
	Quercetin	9.18	151.0	1.14 ± 0.00
	Quercetin 3-O-glucoside	6.10	300.42	0.43 ± 0.02
	Quercetin glucuronide	5.99	300.99	9.32 ± 0.06
	Rutin	5.87	270.94	3.36 ± 0.01
	Catechin	3.60	245.06	10.59 ± 0.04
	Epicatechin	4.42	245.06	0.05 ± 0.00
	Procianidin B1	3.21	289.18	4.56 ± 0.05
	Procianidin B2	1.31	289.00	0.35 ± 0.00
	Epicatechin galate	6.06	256.07	0.01 ± 0.00
**Hydrolysable tannins**	Xyloside ellagic acid	5.28	301.00	0.69 ± 0.12
	Rhamnoside ellagic acid	5.49	300.00	1.27 ± 0.17
	Hexoside ellagic acid	6.62	299.00	18.60 ± 2.14
	Trigalloyl hexoside	3.66	465.00	0.26 ± 0.05
	Castalagin	9.49	458.00	0.05 ± 0.00
	Pedunculagin 1	2.05	246.00	0.02 ± 0.00
	Pedunculagin 2	3.09	245.00	0.03 ± 0.00

Data expressed as mean ± standard deviation.

**Table 3 foods-14-00837-t003:** Parameters of viability, performance, survival, and physicochemical quality of *Akkermansia muciniphila* and *Bifidobacterium longum* microcapsules.

Parameters	Microcapsules *Akkermansia muciniphila*	Microcapsules *Bifidobacterium longum*
Viability (Log 10 UFC/g)	10.44 ± 0.02	10.40 ± 0.04
Drying yield (%)	62.86 ± 1.05	62.25 ± 0.39
Survival percentage (%)	97.02 ± 0.03	98.02 ± 0.39
Water activity (aw)	0.118 ± 0.001	0.117 ± 0.002
Humidity (%)	3.15 ± 0.07	2.85 ± 0.07
Solubility (%)	94.50 ± 0.45	96.77 ± 0.46
Wettability (min)	4.37 ± 0.04	4.11 ± 0.01

Data expressed as mean ± standard deviation.

**Table 4 foods-14-00837-t004:** The survival of microencapsulated *Akkermansia muciniphila* and *Bifidobacterium longum* after in vitro digestion.

Stage	Free Cells of *A. muciniphila* (Log 10 UFC/mL)	Microcapsules *A. muciniphila* (Log 10 UFC/mL)	Free Cells of *B. longum* (Log 10 UFC/mL)	Microcapsules *B. longum* (Log 10 UFC/mL)
Initial	9.00 ± 0.00 ^a^	9.00 ± 0.00 ^a^	9.00 ± 0.00 ^a^	9.00 ± 0.00 ^a^
Oral	8.89 ± 0.13 ^a^	8.99 ± 0.02 ^a^	8.92 ± 0.13 ^a^	8.95 ± 0.16 ^a^
Gastric	6.51 ± 0.24 ^a^	8.97 ± 0.05 ^b^	6.32 ± 0.12 ^a^	8.78 ± 0.16 ^b^
Intestinal	5.86 ± 0.19 ^a^	7.91 ± 0.10 ^b^	5.90 ± 0.30 ^a^	7.77 ± 0.14 ^b^
Viable account decline	3.14 ± 0.19 ^a^	1.09 ± 0.10 ^b^	3.10 ± 0.30 ^a^	1.23 ± 0.14 ^b^

Data expressed as mean ± standard deviation. Significant differences are expressed by different literals according to Tukey (*p* < 0.05).

**Table 5 foods-14-00837-t005:** Nutritional content and physicochemical characteristics of rehydration beverages.

	Formulation of Rehydration Beverages		
Parameter	A (1.6% Agavins)	B (1.2% Agavins)	C (0.8% Agavins)
Humidity	97.65 ± 0.02 ^a^	97.65 ± 0.01 ^a^	97.64 ± 0.02 ^a^
Ashes	0.12 ± 0.00 ^a^	0.12 ± 0.01 ^a^	0.12 ± 0.02 ^a^
Fats	0.00 ± 0.00 ^a^	0.00 ± 0.00 ^a^	0.00 ± 0.00 ^a^
Protein	traces	traces	traces
Carbohydrates	2.23 ± 0.02 ^a^	2.23 ± 0.01 ^a^	2.24 ± 0.02 ^a^
Energy (kcal)	8.92 ± 0.02 ^a^	8.92 ± 0.02 ^a^	8.96 ± 0.02 ^a^
pH	4.11 ± 0.06 ^a^	4.09 ± 0.04 ^a^	4.06 ± 0.05 ^a^
°Brix	2.30 ± 0.00 ^a^	2.30 ± 0.00 ^a^	2.30 ± 0.00 ^a^

Data expressed as mean ± standard deviation. Significant differences are expressed by different literals according to Tukey (*p* < 0.05).

**Table 6 foods-14-00837-t006:** Free profile evaluation of rehydration beverages formulated from IQS, microcapsule probiotics, and prebiotics.

Formulation of Rehydration Beverages					
Beverage Formulation A (1.6% FOS)		Beverage Formulation B (1.2% FOS)		Beverage Formulation C (0.8% FOS)	
Attribute	Frequency	Attribute	Frequency	Attribute	Frequency
Bitter	7	Salty	7	Sour	10
Sour	6	Grass	7	Grass	10
Sweet	4	Bitter	6	Bitter	6
Grass	7	Chamomile	4	Salty	6
Refreshing	3	Sour	4	Dry	3

**Table 7 foods-14-00837-t007:** Phenolic profiling of rehydration beverages formulated from IQS, probiotics, and prebiotics determined by UPLC-ESI-MS/MS, data expressed in μg/mL.

Compound	Formulation			
	A	B	C	IQS
Phenolic acids				
4-hidroxibenzoic acid	0.06 ± 0.00 ^a^	0.04 ± 0.00 ^a^	0.05 ± 0.02 ^a^	0.09 ± 0.00 ^b^
2-hidroxibenozoic acid	0.23 ± 0.05 ^a^	0.15 ± 0.05 ^b^	0.13 ± 0.02 ^b^	0.03 ± 0.00 ^c^
Hidroxibenzoic (derivative)	0.83 ± 0.04 ^a^	0.78 ± 0.04 ^b^	0.55 ± 0.05 ^c^	0.00 ± 0.00 ^d^
t-cinnamic acid	2.63 ± 0.26 ^a^	2.13 ± 0.26 ^b^	1.85 ± 0.37 ^c^	0.02 ± 0.00 ^d^
Protocatechoic acid	3.60 ± 0.22 ^a^	3.18 ± 0.22 ^b^	3.22 ± 0.30 ^c^	1.31 ± 0.00 ^d^
Gallic acid	2.71 ± 0.17 ^a^	3.25 ± 0.39 ^b^	4.61 ± 0.39 ^c^	14.22 ± 0.03 ^d^
Shikimic acid	10.80 ± 0.32 ^a^	21.25 ± 0.32 ^b^	25.50 ± 2.92 ^bc^	28.88 ± 0.02 ^c^
Quinic acid	130.34 ± 2.60 ^a^	127.32 ± 2.60 ^a^	124.48 ± 7.25 ^b^	89.78 ± 0.18 ^c^
Chlorogenic acid	55.73 ± 1.77 ^a^	48.74 ± 1.77 ^b^	43.44 ± 3.94 ^c^	33.04 ± 0.07 ^d^
4-O-caffeoylquinic acid	traces	traces	traces	16.33 ± 0.03
2,5-di-hydroxybenzoic acid	traces	traces	traces	0.11 ± 0.00
Flavonoids				
Catechin	6.49 ± 0.05 ^a^	6.11 ± 0.09 ^b^	5.77 ± 0.44 ^bc^	5.27 ± 0.06 ^c^
Epicatechin	0.03 ± 0.00 ^a^	0.02 ± 0.00 ^a^	0.02 ± 0.00 ^a^	0.02 ± 0.00 ^a^
Kaempferol	traces	traces	traces	0.12 ± 0.00
Kaempferol 3-O-glucoside	0.44 ± 0.01 ^a^	0.19 ± 0.01 ^b^	0.14 ± 0.05 ^bc^	0.12 ± 0.01 ^c^
Quercetin	traces	traces	traces	0.55 ± 0.01
Quercetin 3-O-glucoside	0.86 ± 0.12 ^a^	0.69 ± 0.08 ^b^	0.67 ± 0.03 ^b^	0.21 ± 0.02 ^c^
Quercetin glucuronide	7.07 ± 0.37 ^a^	5.55 ± 0.19 ^b^	5.31 ± 0.23 ^b^	4.68 ± 0.08 ^c^
Rutin	1.02 ± 0.05 ^a^	1.26 ± 0.18 ^b^	1.31 ± 0.06 ^b^	1.68 ± 0.01 ^c^
Phloretin	traces	traces	traces	0.15 ± 0.00
Phloridzin	0.11 ± 0.03 ^a^	0.09 ± 0.01 ^a^	0.08 ± 0.01 ^a^	traces

Data expressed as mean ± standard deviation. Significant differences are expressed by different literals according to Tukey’s test (*p* < 0.05).

**Table 8 foods-14-00837-t008:** Study of stability during 28 days of refrigerated storage (4 °C) for the rehydration drink with the highest FOS content (formulation A).

Time (Days)	Total Phenolics	Total Flavonoids	ABTS	DPPH	ORAC	EMA	*B. longum*	*A. muciniphila*	pH	°Brix
0	8.81 ± 0.11 ^a^	4.48 ± 0.05 ^a^	6.05 ± 0.13 ^a^	7.93 ± 0.07 ^a^	2.32 ± 0.06 ^a^	63.08 ± 0.73 ^a^	6.86 ± 0.09 ^a^	6.82 ± 0.11 ^a^	4.11 ± 0.08 ^a^	2.30 ± 0.00 ^a^
3	9.08 ± 0.09 ^a^	4.63 ± 0.24 ^a^	5.64 ± 0.10 ^b^	7.88 ± 0.10 ^a^	2.42 ± 0.04 ^a^	59.70 ± 1.02 ^a^	6.88 ± 0.12 ^a^	6.92 ± 0.10 ^a^	4.07 ± 0.05 ^a^	2.30 ± 0.00 ^a^
5	8.52 ± 0.07 ^b^	4.38 ± 0.05 ^a^	5.60 ± 0.31 ^b^	7.49 ± 0.21 ^b^	2.60 ± 0.09 ^b^	54.68 ± 1.74 ^b^	6.80 ± 0.10 ^a^	6.82 ± 0.12 ^a^	3.96 ± 0.04 ^b^	2.30 ± 0.00 ^a^
7	8.50 ± 0.17 ^b^	4.54 ± 0.37 ^a^	5.40 ± 0.27 ^b^	7.44 ± 0.21 ^b^	2.44 ± 0.11 ^ab^	52.60 ± 2.14 ^b^	6.62 ± 0.03 ^b^	6.69 ± 0.21 ^a^	3.94 ± 0.02 ^b^	2.30 ± 0.00 ^a^
14	8.52 ± 0.03 ^b^	3.10 ± 0.05 ^b^	4.99 ± 0.42 ^bc^	7.23 ± 0.17 ^b^	1.83 ± 0.23 ^c^	52.55 ± 0.86 ^b^	5.82 ± 0.11 ^c^	6.12 ± 0.24 ^b^	3.89 ± 0.10 ^b^	2.20 ± 0.00 ^b^
28	6.58 ± 0.06 ^c^	3.03 ± 0.04 ^b^	4.84 ± 0.11 ^c^	7.07 ± 0.24 ^c^	1.60 ± 0.12 ^c^	52.60 ± 0.75 ^b^	5.12 ± 0.03 ^d^	4.82 ± 0.23 ^c^	3.88 ± 0.07 ^b^	2.20 ± 0.00 ^b^

Data expressed as mean ± standard deviation. Significant differences expressed by different literals according to Tukey (*p* < 0.05).

## Data Availability

The original contributions presented in this study are included in the article/Appendix A. Further inquiries can be directed to the corresponding author.

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
