# Peer review of "Rehydration Beverages Made from Quercus sideroxyla Infusion, Probiotics, and Prebiotics: Antioxidant and Anti-Inflammatory Potential"

_foods, 2025, doi:10.3390/foods14050837_

Round 1

Reviewer 1 Report

Comments and Suggestions for Authors

Comments to authors:

This manuscript presents an interesting study on the development and evaluation of rehydration beverages formulated with Quercus sideroxyla infusion, probiotics, and prebiotics. The research addresses an important gap in the market by aiming to create a functional beverage with potential health benefits for physically active individuals. The study is well-structured, and the experimental methods are generally appropriate for the research questions. However, there are several areas that could be improved to enhance the quality and impact of the paper.

1 Introduction

In the introduction, it should clearly show the knowledge gaps identified and link them to the paper goals. Please reason both the novelty and the relevance of the paper goals.

While the introduction provides a good overview of the general context, it could be strengthened by including more specific information on the current market share and trends of rehydration beverages. For example, data on the global or regional consumption of rehydration beverages and the market demand for functional beverages with health - promoting ingredients would better justify the need for this study.

2 Materials and Methods

Strain Information: For the probiotic strains Akkermansia muciniphila and Bifidobacterium longum, more information about their origin and characteristics could be provided. For instance, details about their specific physiological functions beyond what is already mentioned, such as their adhesion properties to the intestinal mucosa or their ability to produce specific metabolites, would be valuable.

Quality Control: In the preparation of Quercus sideroxyla leaves infusion, the authors mention that the leaves were collected in a specific location and season. However, there is no information on the quality control measures during collection, such as the assessment of leaf maturity or the presence of contaminants. This information is important to ensure the reproducibility of the results.

3 Results and Discussion

Phenolic Profiling Interpretation: Although the phenolic profiling results are presented in detail, the interpretation of the changes in phenolic compound concentrations in the formulated beverages compared to the Quercus sideroxyla infusion could be more in - depth. For example, the authors could discuss how the complexation between fructooligosaccharides and phenolic compounds affects not only the measured concentrations but also the bioavailability and biological activities of these compounds in vivo.

The detail identification information of phenolic compounds should be provided.

Antioxidant and Anti - Inflammatory Mechanisms: While the results of antioxidant and anti - inflammatory assays are reported, the discussion of the underlying mechanisms could be expanded. For the antioxidant potential, in addition to identifying the key compounds involved, the authors could discuss how these compounds interact with each other and with other components in the beverage to exert their antioxidant effects. Similarly, for the anti - inflammatory potential, more details about how the components, especially fructooligosaccharides, contribute to membrane stabilization at the molecular level would be beneficial.

Sensory Analysis: The sensory analysis results are somewhat limited. The authors could provide more in - depth analysis of the sensory data, such as correlations between different sensory attributes and the chemical composition of the beverages. Additionally, information on how the sensory properties of the beverages might change over time, for example, during storage, would be useful.

In the discussion section, I suggest a deeper exploration of the possible mechanisms behind the observed results. Additionally, it would be beneficial to discuss how these findings relate to and fit into the existing knowledge base in the field. This will provide a more comprehensive understanding of the research and its implications.

4 Conclusions

Generalizability: The conclusions mainly focus on the performance of the formulated beverages in vitro. To enhance the practical significance of the study, the authors should discuss the potential implications of these findings for in - vivo studies and for the target population (physically active individuals). For example, how the antioxidant and anti - inflammatory properties of the beverages might translate into actual health benefits in real - life situations, and what the next steps for further research in this area could be.

5. Please check the references throughout the manuscript.

Comments on the Quality of English Language

The English could be improved to more clearly express the research.

Reviewer 2 Report

Comments and Suggestions for Authors

The authors have presented an experimental  study on the formulation of a rehydrating beverage  made from Quercus sideroxyla leaves, suitable for athletics, particularly individuals who had a sedentary lifestyle and starting out to exercise. The various attributes of this infusion include its anti-oxidant due to the phenolics; and anti-inflammatory properties mediated by the probiotics and prebiotics as well as the postbiotics emanating from them. In addition to its hyadrating power, the beverage is formulated to also supply electrolytes.

  1. Introduction
  • The authors have provided sufficient details covering the benefits of the beverage from Quercus sideroxyla formulated with probiotics and prebiotics. These benefits include improvement of insulin sensitivity, anti-lipidaemic, anti-inflammatory, and anti-oxidant activity, as well as maintenance of mental health.
  • While the properties of Akkermansia muciniphila have been provided, the authors may want to briefly indicate why this organism has been considered instead of members of the Lactobacillus genus, which are commonly paired with Bifidobacterium longum.
  • Authors need to italicize the scientific names of the microorganisms whether written in full or abbreviated.
  1. Materials and methods
  • The authors have provided sufficient details for the study to be repeated elsewhere.
  • The authors need to pay attention to the words highlighted in the reviewed manuscript and correct accordingly.
  • In my view the authors have confused prebiotics and probiotics, this also needs to be corrected.
  • In addition to the other parameters, it would have been good if the authors also measured the Brix of the beverage, and estimated its shelf life or stability, unless alternative measures have been used.
  • I am happy with the statistical treatment of the data, which seems to be logical to me.
  1. Results and Discussion
  • The discussion is sound, following careful data analysis.
  • The authors need to rectify all highlighted areas in this section, particularly remove or replace the Spanish text.
  1. Conclusions
  • Write the scientific names of organisms in italics.

Comments on the Quality of English Language

The English language usage is appropriate, but can be improved by language editor.

Reviewer 3 Report

Comments and Suggestions for Authors

This study investigates rehydration beverages formulated with Quercus sideroxyla infusions, probiotics (Akkermansia muciniphila and Bifidobacterium longum), and prebiotics (FOS from agave).

These are my main questions for Authors:

  1. How does this beverage compare nutritionally to commercial rehydration drinks?
  2. What is the estimated shelf-life of the probiotic formulations?
  3. Would live probiotic counts remain above functional levels after 6–12 months?
  4. How do phenolic-probiotic interactions influence gut microbiota composition?
  5. Could this formulation be modified for endurance athletes with higher electrolyte needs?

Other observations:

The statistical methods are not clearly described for all analyses. No clear indication of p-values, standard deviations, or confidence intervals in key comparisons. The sample size for sensory analysis (n=29) is small for strong conclusions.

The study reports high antioxidant and anti-inflammatory effects but does not explain the molecular mechanisms. The role of specific polyphenols and probiotics in inflammation reduction is not detailed.

The study does not compare its formulations with commercial sports drinks. No benchmarking against leading brands (Gatorade, Powerade, Pedialyte). Maybe to add a comparison table (nutritional profile, bioactivity, sensory attributes) and discuss why this formulation is superior (or not) compared to existing rehydration drinks.

The paper only evaluates short-term stability (90 days). No discussion on storage conditions or shelf-life beyond 3 months.
